# Impaired host resistance to *Salmonella* during helminth co-infection is restored by anthelmintic treatment prior to bacterial challenge

**Tara P. Brosschot, Katherine M. Lawrence, Brandon E. Moeller , Mia H. E. Kennedy ,
Rachael D. FitzPatrick, Courtney M. Gauthier, Dongju Shin , Dominique M. Gatti , Kate
M. E. Conway, Lisa A. Reynolds ***

Department of Biochemistry and Microbiology, University of Victoria, Victoria, Canada

* lisareynolds@uvic.ca

## Abstract

Intestinal helminth infection can impair host resistance to co-infection with enteric bacterial pathogens. However, it is not known whether helminth drug-clearance can restore host resistance to bacterial infection. Using a mouse helminth-*Salmonella* co-infection system, we show that anthelmintic treatment prior to *Salmonella* challenge is sufficient to restore host resistance to *Salmonella*. The presence of the small intestine-dwelling helminth *Heligmosomoides polygyrus* at the point of *Salmonella* infection supports the initial establishment of *Salmonella* in the small intestinal lumen. Interestingly, if helminth drug-clearance is delayed until *Salmonella* has already established in the small intestinal lumen, anthelmintic treatment does not result in complete clearance of *Salmonella*. This suggests that while the presence of helminths supports initial *Salmonella* colonization, helminths are dispensable for *Salmonella* persistence in the host small intestine. These data contribute to the mechanistic understanding of how an ongoing or prior helminth infection can affect pathogenic bacterial colonization and persistence in the mammalian intestine.

## Author summary

In regions where helminth infection is common and sanitation standards are poor, people are at a high risk of exposure to bacterial pathogens. Previous work in animal models has shown that helminth infection can impair host resistance to bacterial infection. The current treatment for helminth infection is the administration of helminth-clearing drugs, yet it is not known whether drug clearance of helminths restores helminth-impaired host resistance to bacterial infection. In this report we use a mouse helminth-*Salmonella* co-infection model system, where we find that the presence of small intestinal helminths at the point of *Salmonella* infection aids the establishment of *Salmonella* in the small intestinal lumen. We show that helminth drug clearance prior to *Salmonella* infection is sufficient to restore host resistance to *Salmonella*. However, if helminth drug clearance is

**Data Availability Statement:** All relevant data are within the manuscript and its Supporting Information files.

**Funding:** This work was supported by a Project Grant to L.A.R. from the Canadian Institutes of Health Research (CIHR; #388288), funds from the University of Victoria, and equipment grants to L.A.R. from the Canadian Foundation for Innovation (CFI; #36419) and the British Columbia Knowledge Development Fund (BCKDF). The funders had no role in study design, data collection and analysis, decision to publish, or preparation of the manuscript.

**Competing interests:** The authors have declared that no competing interests exist.

delayed until after *Salmonella* had already established in the small intestinal lumen, helminth elimination does not result in complete clearance of *Salmonella* from this site. Our work suggests that helminth drug clearance may be beneficial in reducing susceptibility to subsequent intestinal bacterial infections, but that helminth drug clearance after co-infection may not result in clearance of bacterial populations that have firmly established in the intestinal lumen.

## Introduction

Helminths are parasitic worms that cause a significant global health concern [1]: it is estimated that more than one billion people are currently chronically infected with helminths. Anthelmintic treatment, also known as 'deworming', is the current treatment strategy for helminth infection and needs to be periodically administered to at-risk human and livestock populations. Helminth infection has been associated with impaired host resistance to co-infection with various pathogenic microbes, including bacterial pathogens, both in human populations [2–6] and in mouse models of co-infection [7–17]. However, it is not clear whether anthelmintic treatment is sufficient to restore host resistance to microbial pathogens.

We have previously reported that mice infected with the small intestinal helminth *Heligmosomoides polygyrus* are highly susceptible to colonization of the small intestine by the bacterial pathogen *Salmonella enterica* serovar Typhimurium, compared to mice singly infected with *S.* Typhimurium [12]. In this paper, we use the anthelmintic drug pyrantel pamoate to manipulate *H. polygyrus* infection status in mice and examine the resulting effect on host susceptibility to *S.* Typhimurium.

Here, we demonstrate that the presence of *H. polygyrus* is required by *S.* Typhimurium in order to initially establish high levels of colonization in the small intestinal tract, since anthelmintic treatment prior to bacterial challenge restored host resistance to *S.* Typhimurium colonization. We establish that when adult worms are present at the point of *S.* Typhimurium infection *Salmonella* remains largely in the lumen of the small intestine in close association with the adult worms, rather than invading host tissue. Indeed, expression of host tissue invasion genes by *Salmonella* was not required for establishment in the intestine during helminth infection. Additionally, despite anthelmintic treatment prior to bacterial infection being sufficient to restore host resistance to *S.* Typhimurium, we find that once *Salmonella* has established a population in the small intestinal lumen during helminth co-infection anthelmintic treatment does not result in complete clearance of *Salmonella* from the small intestine.

Our findings contribute to the understanding of how concomitant helminth infection affects bacterial pathogens in the intestinal tract. Furthermore, our data suggest that while anthelmintic treatment may reduce opportunities for bacterial pathogens to colonize the mammalian intestinal tract, anthelmintic treatment may not be sufficient to promote clearance of established bacterial pathogens.

## Methods

### Ethics statement

All animal experiments were approved by the University of Victoria's Animal Care Committee and complied with the policies of the Canadian Council on Animal Care.

## Mice

6–13 week old mice were used for all experiments. C57BL/6J, BALB/cJ and eosinophil-deficient ΔdblGATA BALB/cJ mice were initially obtained from The Jackson Laboratory (strain #s 000664; 000651 and 005653 respectively, all from maximum-barrier rooms) and were subsequently bred and maintained under specific-pathogen free conditions at the University of Victoria with access to food and water *ad lib*. Both male and female mice were used for experiments, as indicated in figure legends. For experiments using C57BL/6J mice, pups born to different parents were randomized between treatment groups. For experiments with wild-type BALB/cJ and ΔdblGATA BALB/cJ mice, to minimize the potential effects of microbiota compositional differences between mice of different genotypes on experimental outcomes, female wild-type BALB/cJ and eosinophil-deficient ΔdblGATA BALB/cJ mice were co-housed for at least one week prior to beginning experiments and were kept co-housed throughout the duration of the experiment. Male mice born in different litters could not be co-housed due to fighting, but instead the bedding of male wild-type BALB/cJ and eosinophil-deficient ΔdblGATA BALB/cJ mice was swapped twice weekly starting the week prior to beginning experiments, and throughout the duration of the experiment.

## Infections

The life cycle of *H. polygyrus* was maintained in C57BL/6J mice according to an established protocol [18]. Experimental mice were infected with 200 (unless otherwise indicated) *H. polygyrus* stage 3 larvae by oral gavage. *H. polygyrus* burdens were tracked by counting parasite eggs released into feces, which were enumerated in a McMaster Counting Chamber slide under a light microscope.

All *Salmonella enterica* serovar Typhimurium strains used were streptomycin-resistant (strain SL1344). Mice were infected with *Salmonella* by oral gavage with either $3 \times 10^6$ colony-forming units (cfu) of wild-type *S*. Typhimurium or host-invasion-deficient (Δ*invA*) *S*. Typhimurium [19], or with $3 \times 10^8$ cfu of a growth-attenuated (Δ*aroA*) strain of *S*. Typhimurium [20] as indicated. Inocula were prepared from stationary-phase overnight cultures in Luria-Bertani (LB) broth and were diluted in phosphate-buffered saline (PBS) prior to infection.

## Anthelmintic treatment

Mice were given 2.5 mg Strongid P (Zoetis) in Ultra-Pure Distilled Water (Invitrogen) by oral gavage on two consecutive days. Efficacy of anthelmintic treatment was monitored by tracking fecal *H. polygyrus* egg release. For each experiment we confirmed that fecal *H. polygyrus* egg burdens were equivalent between *H. polygyrus*-infected groups prior to beginning anthelmintic treatment in the dewormed group.

## Streptomycin treatment

For some experiments presented in the Supplementary Information, to deplete the bacterial microbiota, mice received an oral gavage of 20 mg of streptomycin sulfate (GoldBio) diluted in PBS.

## Bacterial burden determination

Serial dilutions of homogenized tissue were plated on LB plates containing 100 μg/mL streptomycin (Sigma-Aldrich) and incubated overnight at 37°C. *Salmonella* colonies were then counted and cfu per gram of tissue was calculated.

To determine *Salmonella* cfu in separated tissue and luminal gut fractions, intestinal sections were cut open longitudinally and luminal contents were scraped out using forceps. Tissue fractions were washed in PBS twice, incubated in RPMI 1640 media supplemented with 100 μg/mL gentamycin (GoldBio) for 45 minutes at room temperature, and then washed in PBS twice. The luminal and tissue fractions were homogenized, plated on streptomycin-containing LB plates and incubated overnight at 37°C, after which cfu per homogenate was calculated.

To determine the proportion of *S*. Typhimurium in association with adult *H. polygyrus* worms, worms were separated from the luminal contents prior to cfu determination. Intestinal sections were cut open longitudinally and luminal contents were scraped out using forceps. Luminal contents (containing worms) were placed in a muslin bag suspended in Hank's Balanced Salt Solution (Gibco) in a Baermann apparatus and incubated at 37°C for 2 hours. Following incubation, the majority of worms had migrated through the muslin bag to the collection funnel, and were subsequently homogenized and plated on streptomycin-containing LB plates (the few remaining worms were removed manually). The luminal contents (with no remaining worms) were recovered, homogenized, and plated on streptomycin-containing LB plates. The tissue fractions were washed in PBS twice, incubated in PBS supplemented with 100 μg/mL gentamycin (GoldBio) for 45 minutes at 37°C, then washed in PBS twice and incubated at 37°C for a further 1 hour 15 mins to be comparable to the incubation time of the worm and luminal content fractions, then homogenized and then plated on streptomycin-containing LB plates. After the plates were incubated overnight at 37°C, cfu per homogenate was calculated.

## Statistical analyses

Statistical analyses were performed in GraphPad Prism 7.04. Normality of the data was assessed by a D'Agostino-Pearson normality test and the appropriate statistical test was performed depending on the normality of the data set, the number of experimental groups being compared, and whether paired data sets were being compared or not, as indicated in the figure legends. A table containing all raw data we used to generate our figures is included in Supporting Information file S1 Table.

## Results

### Deworming prior to bacterial challenge restores host resistance to *Salmonella* in the small intestine

*H. polygyrus* is a natural parasite of mice and is able to establish a chronic infection in the proximal small intestine of C57BL/6J mice [21]. After 14 days of infection with *H. polygyrus*, adult helminths are present in the lumen of the duodenum and jejunum where they wrap around villi to secure their location. We have previously reported that when 14-day *H. polygyrus*-infected mice are challenged with *S*. Typhimurium, *Salmonella* is able to colonize the small intestinal tract to higher levels than when no helminths are present [12]. We find that helminth co-infection enhances *Salmonella* colonization of the small intestine even in mice which lack IL-4, Stat6, and RAG1 [12], and also in mice lacking eosinophils (**S1 Fig**), and bacterial microbiota-depleted mice (**S2 Fig**). Further, we find that there is a relationship between helminth burden and *S*. Typhimurium colonization levels: a higher infectious dose of helminths results in higher *Salmonella* colonization levels in the small intestine (**S3 Fig**). To gain deeper insight into the potential mechanisms by which helminths affect susceptibility to co-infection, we

examined whether the increase in *S.* Typhimurium colonization during helminth infection requires the ongoing presence of helminths.

To test this, once mice had an established *H. polygyrus* infection (day 14 of infection), we treated them with Strongid P (Zoetis). This treatment is widely used for veterinary deworming as it contains the anthelmintic compound pyrantel pamoate (PP). We found that a two-day deworming treatment was sufficient to clear *H. polygyrus* from the mouse intestine, as indicated by the absence of parasite eggs in feces (**Fig 1A**). One day after successful anthelmintic treatment, mice were infected with *S.* Typhimurium, alongside mice that had an ongoing *H. polygyrus* infection and mice with no prior helminth infection (**Fig 1A**). Here, we used a growth-attenuated strain of *S.* Typhimurium (ΔaroA) which allowed our subsequent experiments to explore infection dynamics several days following *Salmonella* infection without mice succumbing to the infection.

Consistent with what we have reported previously [12], those mice who had never been exposed to helminths were able to clear *Salmonella* from the small intestine one day after *Salmonella* infection, whereas *H. polygyrus*-co-infected mice presented with high bacterial burdens (**Fig 1B**). Mice that received deworming treatment prior to *Salmonella* infection had significantly lower bacterial burdens in the small intestine than untreated co-infected mice, and instead, similar to mice that had never been exposed to helminths, were able to clear the majority of *Salmonella* from the small intestine (**Fig 1B**). An ongoing *H. polygyrus* infection and deworming had similar, but less marked effects on *Salmonella* colonization in the large intestine (**Fig 1C**), and helminth infection status did not affect *Salmonella* trafficking to the spleen (**Fig 1D**). We confirmed that there was no effect of pre-treatment with anthelmintics on *Salmonella* colonization levels (**S4 Fig**). Together, these data show that anthelmintic treatment restores host resistance to *Salmonella* in the small intestinal tract within one day of adult worms being cleared, suggesting that an ongoing helminth infection is required to promote *Salmonella* colonization through local and transient (only while *H. polygyrus* is present) alterations to the small intestinal environment.

## The presence of helminths supports luminal expansion of *Salmonella*

Local changes during *H. polygyrus* infection include significant shifts in the availability of metabolites in the small intestine [12]. The composition of metabolites in the intestinal tract can alter the ability of *Salmonella* to colonize the intestinal tract through multiple mechanisms, for example, by affecting the availability of nutrients or by influencing *Salmonella* virulence gene expression [22]. We have previously shown that small intestinal metabolites from naïve mice can suppress *S.* Typhimurium genes required for host tissue invasion in *in vitro* assays, whereas small intestinal metabolites from *H. polygyrus*-infected mice lack this effect [12]. Therefore, we asked if *S.* Typhimurium was taking advantage of a helminth-altered environment that promotes *Salmonella* invasion of small intestinal tissue.

To test this, mice were co-infected with *H. polygyrus* and an invasion-deficient *S.* Typhimurium mutant (Δ*invA*) [19] (**Fig 2A**). InvA expression is required to form a type 3 secretion system which allows *S.* Typhimurium to invade host cells [19]. We found that colonization of the small intestine by Δ*invA S.* Typhimurium was promoted by the presence of helminths to the same extent as wild-type *S.* Typhimurium (**Fig 2B**). This suggests that the helminth-modified small intestinal environment does not result in increased host tissue invasion by *Salmonella*.

To further support our conclusion that increased bacterial host tissue invasion does not underlie increased *Salmonella* colonization during helminth infection, we tested whether *S.* Typhimurium predominantly colonizes the intestinal lumen, rather than host tissue, during

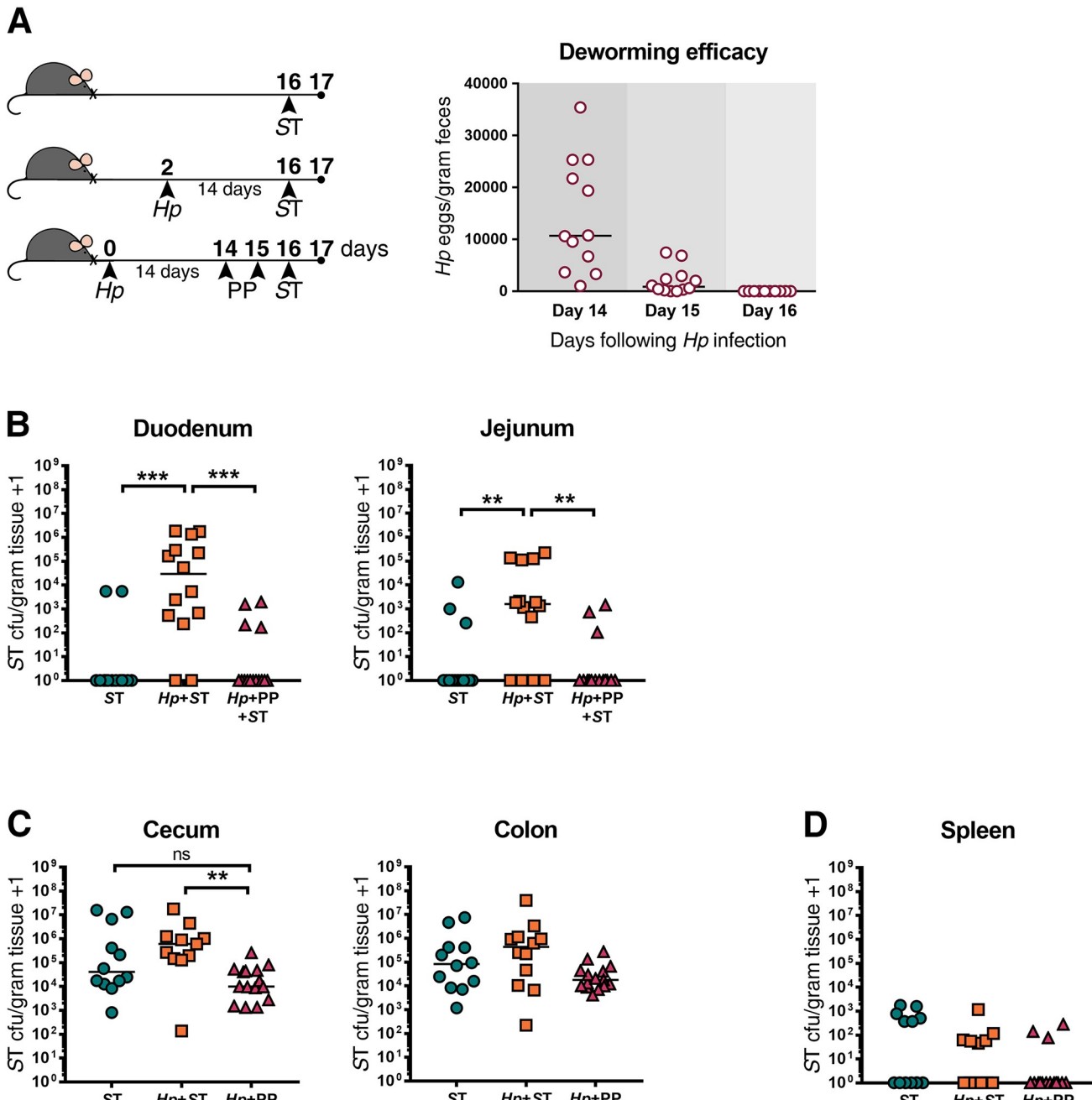

**Fig 1. Deworming prior to bacterial challenge restores host resistance to *Salmonella* Typhimurium (*ST*) in the small intestine of mice.** (A) Experimental set-up and deworming efficacy. Naïve or *H. polygyrus* (*Hp*)-infected male and female C57BL/6J mice were orally infected with Δ*aroA ST* or given an oral dose of deworming drug (PP) for two consecutive days, fourteen days post *Hp*-infection. Mice that received PP were subsequently orally infected with *ST*. One day post-*ST* infection, *ST* colony-forming units (cfu)/gram of tissue were determined. Numbers of *Hp* eggs released in feces were quantified on days 14–16 from mice receiving deworming treatment, as a non-terminal method of assessing worm burdens, to confirm anthelmintic treatment efficacy. *ST* cfu/gram of tissue in the duodenum and jejunum (B), cecum and colon (C), and spleen (D) are shown. Data shown are pooled from three independent experiments. Statistical comparisons between groups were made using a Kruskal-Wallis test followed by a Dunn's multiple comparisons test. A line indicates the median value for each experimental group. ns = not significant; ** = p $\leq$ 0.01; *** = p $\leq$ 0.001.

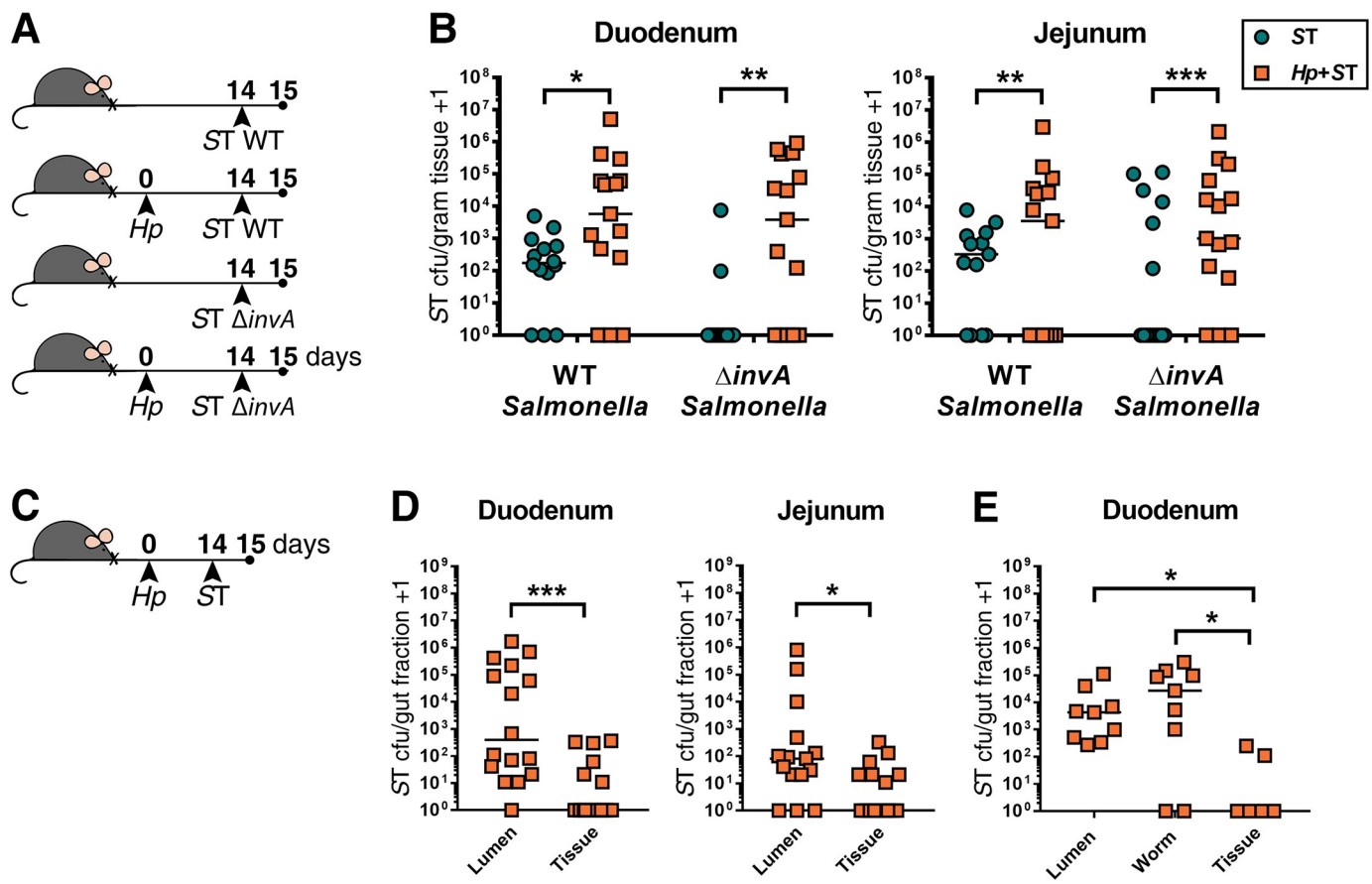

**Fig 2. *Salmonella* Typhimurium (*ST*) predominantly expands in the small intestinal lumen during helminth co-infection.** (A) Experimental set-up. Naïve or *H. polygyrus* (*Hp*)-infected male and female C57BL6/J mice were orally infected with wild-type *ST* (WT *ST*) or invasion-deficient ST (Δ*invA* *S*T) fourteen days post-*Hp* infection. One day post-*S*T infection, *S*T colony-forming unit (cfu) counts were determined in the duodenum and jejunum (B). Data shown are pooled from three independent experiments. Statistical comparisons between singly- and co-infected mice infected with either WT or Δ*invA* *S*T were calculated with a Mann-Whitney test. A line indicates the median value for each experimental group. (C) Experimental set-up. Male and female C57BL6/J mice were infected with *H. polygyrus* (*Hp*). Fourteen days post-*Hp* infection, mice were orally infected with Δ*aroA* *S*T. One day post-*S*T infection, small intestinal sections were dissected and *S*T cfu were determined in the duodenum and jejunum. (D) *S*T cfu in luminal and tissue small intestinal fractions. Data shown are pooled from three independent experiments. Statistical comparisons between groups were made using a Wilcoxon matched-pairs signed-rank test. (E) *S*T cfu in luminal fraction with adult worms removed, in extracted *Hp* worm fraction, and in tissue fractions of the duodenum. Data shown are pooled from two independent experiments. Statistical comparisons between groups were made using a Friedman test followed by a Dunn's multiple comparisons test. A line indicates the median value for each experimental group. * = p ≤ 0.05; ** = p ≤ 0.01; *** = p ≤ 0.001.

helminth co-infection. Small intestinal luminal contents were separated from the intestinal tissue of helminth co-infected mice one day after bacterial infection and *Salmonella* burdens were quantified in both fractions (**Fig 2C**). We found that during co-infection *Salmonella* is present in significantly higher numbers in the luminal fraction than in the small intestinal tissue (**Fig 2D**). Notably, many of these bacteria were found in close association with adult *H. polygyrus* worms isolated from the luminal contents (**Fig 2E**). Quantification of bacterial burdens nine days following *S*. Typhimurium infection revealed that *Salmonella* persists in the lumen of the small intestine during helminth co-infection for an extended period of time (**S5 Fig**). Based on our collective data, we suggest that *H. polygyrus* enables *Salmonella* to overcome host resistance to colonization and to expand primarily in the small intestinal lumen, and, to a lesser extent, in the intestinal tissue.

### *Salmonella* does not require the ongoing presence of helminths to persist in the small intestine

It is possible that after overcoming host resistance to colonization during helminth co-infection, *Salmonella* evades or overwhelms local immune defenses which enables long-term persistence of the bacteria. Therefore, we asked whether *S.* Typhimurium required the ongoing presence of worms in order to persist in the small intestine. To test this, we assessed whether resistance to *Salmonella* could be restored by deworming once *Salmonella* had co-colonized with helminths. We first confirmed that anthelmintic treatment itself had no effect on *Salmonella* persistence, when given after *Salmonella* infection (S6 Fig). Mice were co-infected with *H. polygyrus* and ΔaroA *S.* Typhimurium and then anthelmintic-treated one day after *Salmonella* infection, alongside singly *Salmonella*-infected and co-infected mice that did not receive anthelmintic treatment (Fig 3A). We confirmed that helminths were cleared within 24 hours of the last dose of anthelmintics (S7 Fig) and then quantified *Salmonella* burdens 24 hours later (Fig 3B–3D). In mice that received deworming treatment following co-infection, a subset of *S.* Typhimurium persisted in the duodenum, resulting in bacterial burdens that did not significantly differ from levels detected in mice with an ongoing helminth infection (Fig 3B). To

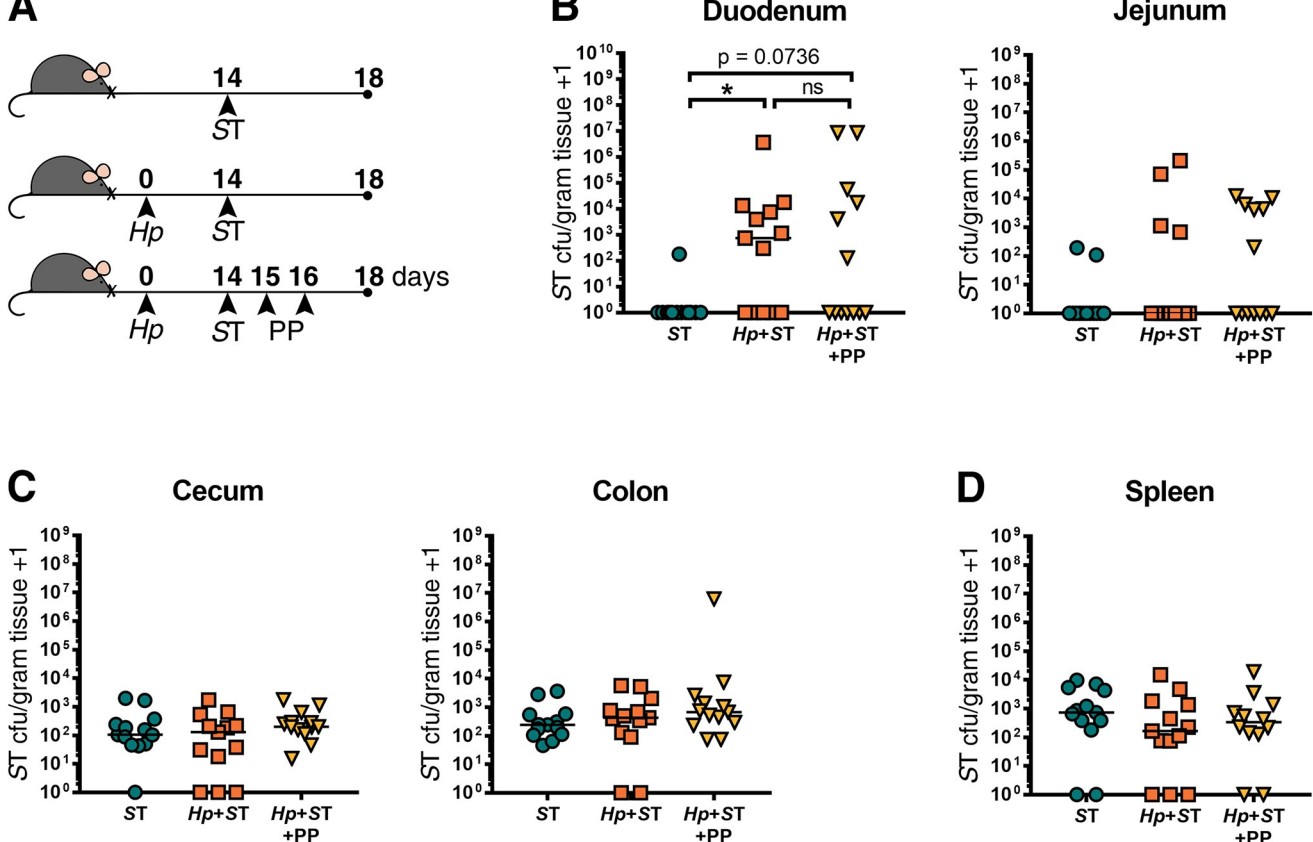

**Fig 3. A subset of *Salmonella* Typhimurium (ST) persists in the small intestine 24 hours after helminth clearance.** (A) Experimental set-up. Naïve or *H. polygyrus* (*Hp*)-infected male and female C57BL/6J mice were orally infected with Δ*aroA S*T fourteen days post-*Hp* infection. One day post-*S*T infection, helminth-co-infected mice were given deworming treatment (PP) for two days or not. One day following PP treatment or no treatment, *S*T colony-forming units (cfu)/gram of tissue were determined. *S*T cfu/gram of tissue in the duodenum and jejunum (B), cecum and colon (C), and the spleen (D) are shown. Data shown are pooled from two independent experiments. Statistical comparisons between groups were made with a Kruskal-Wallis test followed by a Dunn's multiple comparisons test. A line indicates the median value for each experimental group. ns = not significant; * = p ≤ 0.05.

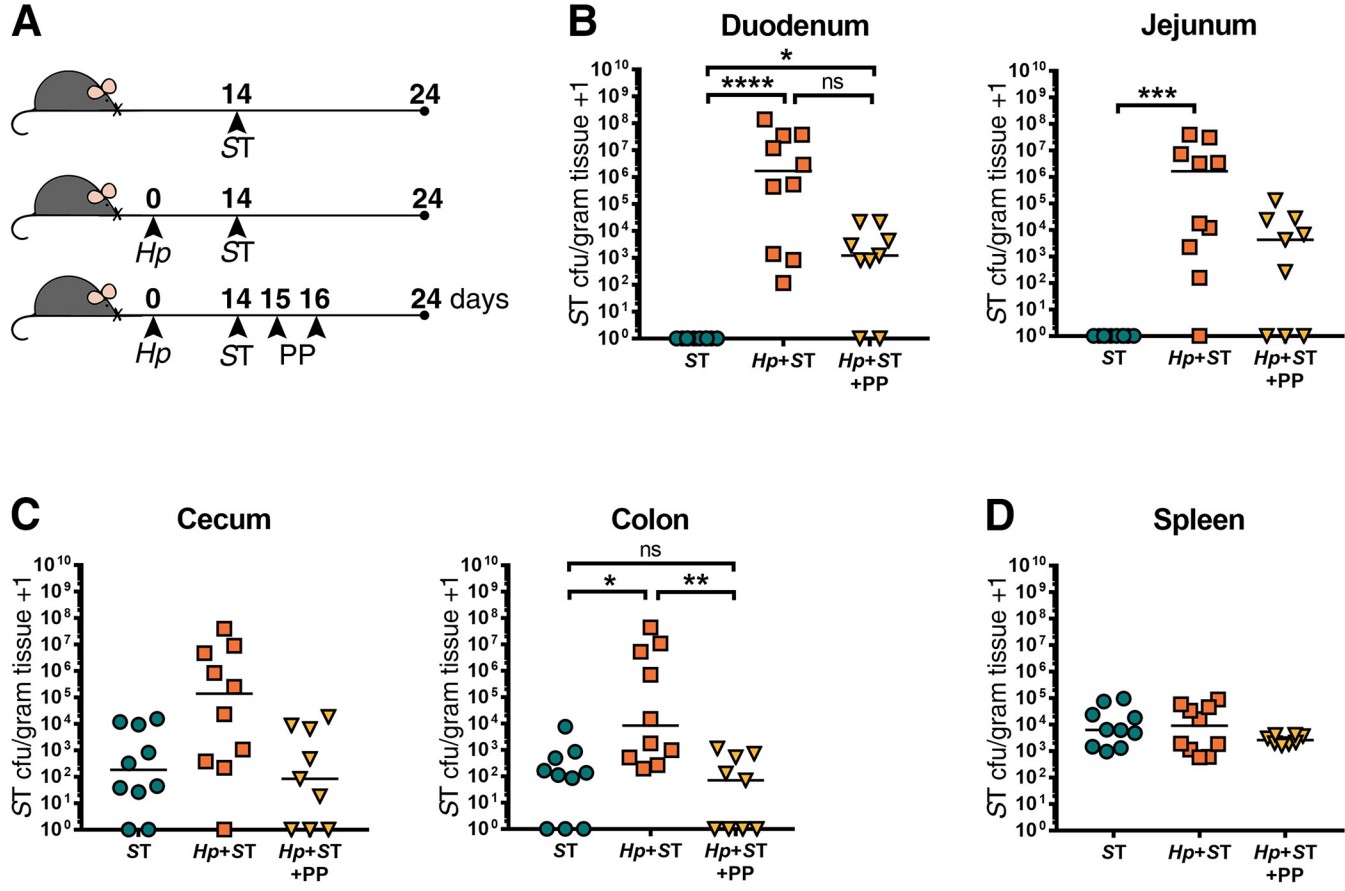

**Fig 4. *Salmonella* Typhimurium (ST) is able to persist in the small intestine one week after helminth clearance.** (A) Experimental set-up. Naïve or *H. polygyrus* (*Hp*)-infected female C57BL/6J mice were orally infected with Δ*aroA S*T fourteen days post-*Hp* infection. One day post-*ST* infection, *Hp*-co-infected mice were given deworming treatment (PP) for two days or not. Eight days post-treatment or no treatment, *ST* colony-forming units (cfu)/gram of tissue were determined in all groups. *ST* cfu/gram of tissue in the duodenum and jejunum (B), cecum and colon (C), and the spleen (D) are shown. Data shown are pooled from two independent experiments. Statistical comparisons between groups were made using a Kruskal-Wallis test followed by a Dunn's multiple comparisons test. A line indicates the median value for each experimental group. ns = not significant; * = p ≤ 0.05; ** = p ≤ 0.01; *** = p ≤ 0.001; **** = p ≤ 0.0001.

further resolve differences in *Salmonella* burdens between experimental groups, we followed the same experimental timeline but used the faster-growing, wild-type *S.* Typhimurium strain for infections, and confirmed that deworming following helminth-*Salmonella* co-infection did result in significantly lower small intestinal *Salmonella* burdens compared to mice with an ongoing helminth infection (**S8 Fig**).

To determine whether *Salmonella* persisting in the small intestine following helminth clearance would be ultimately cleared from this site, we followed the same experimental protocol as in (**Fig 3A**), but with quantification of Δ*aroA S.* Typhimurium burdens at a later timepoint: one week after we had confirmed helminth clearance (**Figs 4A and S7**). One week following helminth clearance, we found that *Salmonella* was still detectable in the small intestine, with significantly higher *Salmonella* burdens in the small intestine than in non-helminth-infected controls, and with *Salmonella* burdens that did not significantly differ from the *Salmonella* burdens of mice with an ongoing helminth infection (**Fig 4B**). At this later time point, we saw higher *Salmonella* burdens in the colon of helminth co-infected mice compared to mice singly infected with *Salmonella*, which did not persist when mice received deworming treatment (**Fig 4C**). This suggests that deworming reduced overall *Salmonella* burdens despite a subset of

bacteria persisting in the small intestine. We found that it was specifically the luminal fraction of *Salmonella*, rather than those bacteria that had invaded the host small intestinal tissue, that was reduced after deworming (**S9 Fig**). We found no differences in systemic dissemination of *Salmonella* regardless of helminth infection status (**Figs 3D, 4D and S8D**). Overall, these results show that once *Salmonella* has established in the small intestine during helminth co-infection, the presence of the worm is no longer essential for *Salmonella* to persist in the small intestine, but that anthelmintic treatment does reduce *Salmonella* colonization levels.

## Discussion

In this paper, we investigated the changes in host resistance to a pathogenic bacterial infection when helminth-infected hosts received anthelmintic treatment either before or after bacterial challenge. We demonstrated that helminth-infected mice that received deworming treatment prior to *S.* Typhimurium infection showed restored colonization resistance to *Salmonella* in the small intestine, within a day of helminth clearance. Based on this finding, we suggest that the benefits of mass deworming in helminth-endemic areas may extend to attenuating gastro-intestinal colonization by bacterial pathogens such as *Salmonella*. The effect of anthelmintic treatment prior to challenge with microbial pathogens has been investigated in other helminth-microbial co-infection systems with similar effects. For example, in a mouse model of trematode-*Plasmodium* co-infection, exacerbation of *Plasmodium* parasitaemia due to intestinal trematode infection was reversible by deworming when mice were dewormed prior to *Plasmodium* infection [8].

Our data suggests that the effect of deworming on host susceptibility to microbial pathogens is influenced by whether deworming occurs prior to infection with microbial pathogens, or after a microbial infection has already established in the presence of helminths. In this study we found that anthelmintic treatment of helminth-*Salmonella* co-infected mice did not result in complete elimination of *Salmonella* from the small intestine one week after helminths had been cleared. These data may help to explain observations from human population studies which reported that deworming was not associated with improved tuberculosis outcome in a human population [23] and that deworming of hookworm-HIV co-infected patients was not associated with improved T cell counts [24,25]. However, when HIV patients were co-infected with the roundworm *Ascaris lumbricoides*, deworming was associated with improved T cell counts [24,25]. A study on helminth-infected African buffalo found that anthelmintic treatment did not affect the risk of acquiring a bovine tuberculosis infection, but did find that anthelmintic improved survival rates following acquisition of tuberculosis [26]. The impact of helminth colonization and subsequent deworming on host immunity to secondary pathogens likely depends on a multitude of factors, including the particular species of helminths and microbial pathogen, the route of infection, as well as host genetic and environmental factors. Helminth species differ in their niche(s) within their host, their mechanisms of interaction with the host immune system, and their effects on the intestinal microbiota [27–29], and thus the impact of helminth co-infection on host susceptibility to secondary pathogens is likely highly species- and context-dependent. Our work provides an example of a helminth species that can enhance intestinal colonization of a bacterial pathogen, yet it should be noted that in other contexts helminth species have been reported to provide protection against microbial pathogens, particularly when microbial challenge occurred at non-intestinal sites [30–33].

We have shown that helminth infection aids in the initial colonization of *Salmonella*, but that once *S.* Typhimurium has co-colonized with helminths, *Salmonella* does not require the presence of helminths in order for a subset of bacteria to persist in the small intestine. Recently, it was shown that diet-induced microbiota perturbation enhanced the ability of *S.*

Typhimurium to colonize the intestinal tract, and reverting back to the original diet after *S.* Typhimurium establishment did not result in *Salmonella* clearance [34]. This supports the hypothesis that once an opportunity arises for *S.* Typhimurium to initially establish in the intestinal tract, it may no longer require the environmental stimulus that supported colonization in order to persist.

Our current understanding of helminth-bacterial co-infections lacks information on the mechanism(s) by which certain pathogenic bacteria acquire a colonization advantage during helminth infection. Some studies have attributed this to the ability of helminths to modulate immune responses that lower host defense to other bacterial pathogens [9,12,14,16,17]. Another possible mechanism by which helminths can promote bacterial infection is through shifts in the gut metabolic environment. It is known that intestinal metabolites derived from the bacterial microbiota can contribute to colonization resistance to pathogenic bacteria [22]. For example, butyrate can inhibit *Salmonella* pathogenicity island 1 (SPI-1) expression, a gene cluster which enables *Salmonella* to invade host tissue [35]. We have previously found that *H. polygyrus* infection alters the composition of metabolites in the small intestine, and that *H. polygyrus*-modified small intestinal metabolites are unable to inhibit SPI-1 gene expression *in vitro*, in contrast to metabolites from the small intestine of naïve mice which suppress SPI-1 gene expression [12]. Despite this finding, we have demonstrated that *Salmonella* can take advantage of the helminth-modified gut environment even when *Salmonella* lacked the ability to invade host tissue, suggesting that mechanisms beyond the promotion of *Salmonella* host invasion support small intestinal colonization. In fact, we found that the primary location for *Salmonella* expansion during helminth infection was in the small intestinal lumen, and that a portion of *Salmonella* was in close proximity with adult *H. polygyrus* worms. It remains possible that shifts in metabolite availability during helminth infection support *S.* Typhimurium colonization. A previous report has described how gut inflammation can promote luminal expansion of *S.* Typhimurium due to newly available metabolic substrates, which cause a switch to bacterial aerobic respiration [22].

Eosinophilia is a hallmark of helminth infection, which we hypothesized contributed to intestinal inflammation to indirectly create a favourable environment for *Salmonella* colonization. However, using eosinophil-deficient mice, we found that *H. polygyrus* co-infection supported *Salmonella* colonization independently of the presence of eosinophils. Previously, we have shown that helminth induction of IL-4-, Stat6- and RAG1- dependent immune responses are also not essential for helminth-induced *Salmonella* colonization [12]. Additionally, we tested the hypothesis that the presence of an intact intestinal bacterial microbiota was essential for *Salmonella* to benefit from helminth infection, since helminth-induced changes in the microbiota composition have been associated with changes in host immunity [28]. We found that depletion of the bacterial microbiota does not preclude the ability of helminths to promote *Salmonella* colonization in the small intestine.

Our data showing that *H. polygyrus* promotes expansion of *Salmonella* predominantly in the small intestinal lumen may point to a direct interaction between helminths and *Salmonella*. Helminth infection may promote luminal *Salmonella* expansion through providing a favourable attachment surface. It has previously been reported that *S.* Typhimurium evades antibiotic lethality through intimate binding to flatworms [36]. Our data support the hypothesis that *Salmonella* closely associates with worms to promote its initial establishment in the small intestine. However, based on our data obtained through anthelmintic treatment following *Salmonella* colonization, we suggest that the worm surface is no longer essential for *Salmonella* to persist in the small intestine once helminths are drug-cleared.

Together, our findings suggest that anthelmintic treatment may be beneficial in preventing new colonization events with potential enteric bacterial pathogens. Further, anthelmintic

treatment given subsequent to helminth-bacterial co-infection may serve to reduce bacterial burdens. However, anthelmintic treatment may not necessarily result in complete clearance of established bacterial pathogens. These results contribute to the growing literature on the interplay between helminths and co-infecting microbial pathogens and emphasize the importance of understanding the immunomodulatory effects of particular helminth species both during an ongoing infection and following helminth clearance.

## Supporting information

**S1 Table. Raw data used to generate all figures in this publication.**
(XLSX)

**S1 Fig. Eosinophil deficiency does not preclude the ability of helminths to promote *Salmonella* Typhimurium (*S*T) colonization in the small intestines.** (A) Experimental set-up. Naïve or *H. polygyrus* (*Hp*)-infected wild-type and eosinophil-deficient Δ*dblGATA* BALB/cJ mice were orally infected with Δ*aroA S*T fourteen days post *Hp*-infection. One day post-*S*T infection, *S*T colony-forming units (cfu)/gram of tissue were determined. *S*T cfu/gram of tissue in the duodenum (B), jejunum (C), cecum (D), colon (E), and spleen (F) are shown. Data shown are pooled from three independent experiments including both male and female mice. Statistical comparisons for each mouse genotype were made using a Mann-Whitney test. A line indicates the median value for each experimental group. $^{*}$ = p $\leq$ 0.05 $^{**}$ = p $\leq$ 0.01; $^{****}$ = p $\leq$ 0.0001.
(TIF)

**S2 Fig. Bacterial microbiota depletion does not preclude the ability of helminths to promote *Salmonella* Typhimurium (*S*T) colonization in the small intestines.** (A) Experimental set-up. Naïve or *H. polygyrus* (*Hp*)-infected male and female C57BL/6J mice were given a 20 mg dose of streptomycin by oral gavage thirteen days post *Hp*-infection, or left untreated. One day following treatment, all mice were infected with Δ*aroA S*T. One day post-*S*T infection, *S*T colony-forming units (cfu)/gram of tissue were determined. *S*T cfu/gram of tissue in the duodenum and jejunum (B), cecum and colon (C), and spleen (D) are shown. Data shown are pooled from two independent experiments. Statistical comparisons between groups were made using a Kruskal-Wallis test followed by a Dunn's multiple comparisons test. A line indicates the median value for each experimental group. $^{*}$ = p $\leq$ 0.05; $^{**}$ = p $\leq$ 0.01; $^{***}$ = p $\leq$ 0.001; $^{****}$ = p $\leq$ 0.0001.
(TIF)

**S3 Fig. A threshold dose of *Heligmosomoides polygyrus* (*Hp*) is required for a loss of host resistance to *Salmonella* Typhimurium (*S*T) in the small intestine.** Female C57BL/6J mice were left naïve ('0L3') or infected with 25 ('25L3'), 100 ('100L3'), or 200 ('200L3') third stage *H. polygyrus* larvae (L3). Fourteen days later, mice were orally infected with Δ*aroA S*T. One day post-*S*T infection, *S*T colony-forming units (cfu)/gram of tissue were determined in the duodenum. Data shown are pooled from two independent experiments. Statistical comparisons between groups were made using a Kruskal-Wallis test followed by a Dunn's multiple comparisons test. A line indicates the median value for each experimental group. $^{*}$ = p $\leq$ 0.05; $^{***}$ = p $\leq$ 0.001.
(TIF)

**S4 Fig. *Salmonella* Typhimurium (*S*T) colonization is not affected by pre-treatment with a two-day oral dose of 2.5 mg dose of the anthelmintic drug Strongid P (PP) prior to *S*T inoculation.** (A) Experimental set-up. Naïve male and female C57BL/6J mice were treated

with 2.5 mg Strongid P for two consecutive days or left untreated. One day after completion of PP treatment, mice were infected with wild-type *ST* to assess *ST* colony forming units (cfu) in the duodenum (B), or mice were infected with Δ*aroA* *S*T to assess *ST* cfu in the colon (C) and spleen (D). We used wild-type *ST* to assess the effect of PP on *ST* in the duodenum, rather than Δ*aroA* *S*T, because Δ*aroA* *S*T establishes at only low levels in the small intestine in the absence of helminths, making it impossible to detect a potentially adverse effect of PP on *ST* colonization in the small intestines. Wild-type *ST* is able to establish sufficiently in the small intestine, which allows us to determine whether PP affects colonization levels of *ST*. Data shown in both (B) and (C+D) are pooled from two independent experiments, and statistical comparisons between groups were made using a Mann-Whitney test. A line indicates the median value for each experimental group. ns = not significant.
(TIF)

**S5 Fig.** *Salmonella* **Typhimurium (*ST*) persists in the lumen of the small intestine during helminth infection for 9 days following *ST* co-infection.** (A) Experimental set-up. Naïve or *H. polygyrus* (*Hp*)-infected male and female C57BL/6J mice were orally infected with Δ*aroA* *ST* fourteen days post *Hp*-infection. Nine days post-*ST* infection, *ST* colony-forming units (cfu)/gram of tissue were determined. *ST* cfu/gram of tissue in the duodenum and jejunum (B), cecum and colon (C), and the spleen (D) are shown. Data shown are pooled from three independent experiments. Statistical comparisons between groups were made using a Mann-Whitney test. In a different set of experiments following the same experimental timeline, the duodenum and jejunum were dissected to separate out tissue and luminal fractions, and *ST* colony-forming units (cfu) were determined in each fraction (E). Data shown are pooled from three independent experiments. Statistical comparisons between groups were made using a Wilcoxon matched-pairs signed rank test. A line indicates the median value for each experimental group. * = p ≤ 0.05 ** = p ≤ 0.01; **** = p ≤ 0.0001.
(TIF)

**S6 Fig.** *Salmonella* **Typhimurium (*ST*) colonization levels are not affected by a two-day oral dose of 2.5 mg dose Strongid P (PP) following ST inoculation.** Male C57BL/6J mice were infected with wild-type *ST*. On the first and second day after *ST* infection, mice were treated with a 2.5 mg dose of Strongid P or left untreated. One day after completion of PP treatment, *ST* colony-forming unit (cfu) counts were determined in the duodenum. The purpose of this experiment was to test whether PP treatment had any effect on *ST* colonization levels in the small intestine. Because we were looking to detect a potential reduction in *ST* burdens following PP treatment, we used wild-type *ST* rather than Δ*aroA* *S*T, since wild-type *ST* colonizes to higher levels in the small intestine which would allow us to detect a potential reduction in colonization after PP treatment. Data shown are pooled from two independent experiments. Statistical comparisons between groups were made using a Mann-Whitney test. A line indicates the median value for each experimental group. ns = not significant.
(TIF)

**S7 Fig. A two-day deworming treatment is sufficient for** *Heligmosomoides polygyrus* **(*Hp*) clearance.** Numbers of *Hp* eggs released in feces were quantified on days 14–16 from mice receiving deworming treatment, as a non-terminal method of assessing worm burdens, to confirm anthelmintic treatment efficacy.
(TIF)

**S8 Fig. Deworming results in reduced wild-type** *Salmonella* **Typhimurium (*ST*) burdens in the small intestine compared to mice with an ongoing helminth infection.** (A) Experimental set-up. Naïve or *H. polygyrus* (*Hp*)-infected female C57BL/6J mice were orally infected with

wild-type *S*T fourteen days post-*Hp* infection. One day post-*S*T infection, *Hp*-co-infected mice were given deworming treatment (PP) for two days or not. Two days post-treatment or no treatment, *S*T colony-forming units (cfu)/gram of tissue were determined in all groups. *S*T cfu/gram of tissue in the duodenum and jejunum (B), cecum and colon (C), and the spleen (D) are shown. Data shown are pooled from two independent experiments. Statistical comparisons between groups were made using a Kruskal-Wallis test followed by a Dunn's multiple comparisons test. A line indicates the median value for each experimental group. ns = not significant; * = p ≤ 0.05; *** = p ≤ 0.001; **** = p ≤ 0.0001.
(TIF)

**S9 Fig. Deworming after *S*. Typhimurium (*S*T) has colonized during helminth infection leads to a reduction in luminal but not tissue-resident *S*T burdens.** (A) Experimental set-up. Male and female C57BL6/J mice were infected with *H. polygyrus* (*Hp*). Fourteen days post-*Hp* infection, mice were orally infected with Δ*aroA S*T. One day post-*S*T infection, *Hp*-co-infected mice were given deworming treatment (PP) for two days or not. Eight days post-treatment or no treatment, *S*T colony-forming units (cfu)/gram of tissue were determined. (B) *S*T cfu in luminal and tissue small intestinal fractions. Data shown are pooled from two independent experiments. Statistical comparisons were made between the indicated groups using a Mann-Whitney test. A line indicates the median value for each experimental group. * = p ≤ 0.05.
(TIF)

## Acknowledgments

We would like to thank Dr. Bruce A. Vallance for providing us with SL1344 Δ*invA S.* Typhimurium, and Dr. Lisa C. Osborne for assistance and discussions which contributed to this project.

## Author Contributions

**Conceptualization:** Tara P. Brosschot, Lisa A. Reynolds.

**Data curation:** Tara P. Brosschot, Lisa A. Reynolds.

**Formal analysis:** Tara P. Brosschot, Katherine M. Lawrence, Lisa A. Reynolds.

**Funding acquisition:** Lisa A. Reynolds.

**Investigation:** Tara P. Brosschot, Katherine M. Lawrence, Brandon E. Moeller, Mia H. E. Kennedy, Rachael D. FitzPatrick, Courtney M. Gauthier, Dongju Shin, Dominique M. Gatti, Kate M. E. Conway, Lisa A. Reynolds.

**Methodology:** Tara P. Brosschot, Lisa A. Reynolds.

**Project administration:** Tara P. Brosschot, Lisa A. Reynolds.

**Resources:** Lisa A. Reynolds.

**Supervision:** Tara P. Brosschot, Lisa A. Reynolds.

**Validation:** Tara P. Brosschot, Katherine M. Lawrence, Lisa A. Reynolds.

**Visualization:** Tara P. Brosschot, Lisa A. Reynolds.

**Writing – original draft:** Tara P. Brosschot, Lisa A. Reynolds.

**Writing – review & editing:** Tara P. Brosschot, Katherine M. Lawrence, Brandon E. Moeller, Mia H. E. Kennedy, Rachael D. FitzPatrick, Courtney M. Gauthier, Dongju Shin, Dominique M. Gatti, Kate M. E. Conway, Lisa A. Reynolds.

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
