## [Decision Letter · Decision Letter 0]

24 Aug 2020

Dear Dr Reynolds,

Thank you very much for submitting your manuscript "Impaired host resistance to Salmonella during helminth co-infection is restored by anthelmintic treatment prior to bacterial challenge" for consideration at PLOS Neglected Tropical Diseases. As with all papers reviewed by the journal, your manuscript was reviewed by members of the editorial board and by several independent reviewers. In light of the reviews (below this email), we would like to invite the resubmission of a significantly-revised version that takes into account the reviewers' comments. 

We cannot make any decision about publication until we have seen the revised manuscript and your response to the reviewers' comments. Your revised manuscript is also likely to be sent to reviewers for further evaluation.

Sincerely,

Subash Babu

Guest Editor

Christine Petersen

Deputy Editor

Reviewer's Responses to Questions

**Key Review Criteria Required for Acceptance?**

**Methods**

-Are the objectives of the study clearly articulated with a clear testable hypothesis stated?

-Is the study design appropriate to address the stated objectives?

-Is the population clearly described and appropriate for the hypothesis being tested?

-Is the sample size sufficient to ensure adequate power to address the hypothesis being tested?

-Were correct statistical analysis used to support conclusions?

-Are there concerns about ethical or regulatory requirements being met?

Reviewer #1: yes

Reviewer #2: The authors provide a rigorous and clear study design with appropriate controls in the main and supplemental figures, and sample size was appropriate.

**Results**

-Does the analysis presented match the analysis plan?

-Are the results clearly and completely presented?

-Are the figures (Tables, Images) of sufficient quality for clarity?

Reviewer #1: yes

Reviewer #2: For the most part: Results are clearly presented and easy to follow. Quality of the graphs are high.

Some validation data that are lacking in the deworming figures are fecal egg burdens of untreated vs PP-treated mice:

- In fig 1, it seems the data is not shown for day 15 and 16 of Hp alone (unless those mice cleared Hp?). This is a necessary control.

- In Fig 4, egg burden data for Hp and Hp+PP is needed.

**Conclusions**

-Are the conclusions supported by the data presented?

-Are the limitations of analysis clearly described?

-Do the authors discuss how these data can be helpful to advance our understanding of the topic under study?

-Is public health relevance addressed?

Reviewer #1: yes

Reviewer #2: The conclusion, that H.polygyrus impaired resistance to Salmonella requires ongoing H.polygyrus infection during bacterial inoculation, is supported by the experiments administering deworming PP before or after inoculation. These findings are of public health relevance for co-infections and understanding the consequences of pre-existing helminth infections for subsequent bacterial infections.

The authors address another point about the necessity for invasive Salmonella for this phenotype, however, these data are not complete to allow conclusions to be drawn. Required experiments would be to test invA mutant Salmonella in the experimental design for PP-deworming after bacterial inoculation.

**Editorial and Data Presentation Modifications?**

Reviewer #1: (No Response)

Reviewer #2: - Data is presented clearly and in a well-organized manner that makes it easy to follow.

- The relevance of figure 3 is unclear, and should be added to another figure, or in supplemental. Alternatively, this distribution should be considered in the context of the deworming treatments, or the use of the invA mutant.

- I'd advise that Fig S5 be included

**Summary and General Comments**

Reviewer #1: The premise and theme of this work are important; how are helminths and bacteria interacting in the mammalian gut and what is the effect of anthelminthic treatment on colonization of the host with pathogenic bacteria.

The manuscript is logical and clearly written. The data are clearly presented, with the appropriate statistical analyses and the conclusions are not over-stated. The discussion is long and could be cut by 25% (there is some redundancy in sentence structure).

The authors have reproduced some of their previously published data (i.e. infection with H. polygyrus increases Salmonella colonization). Here they extend this by showing that: (1) clearance of H. polygyrus with an anthelminthic treatment prevents the enhanced colonization when Salmonella are inoculated; (2) the enhanced colonization by Salmonella is independent of the bacterium’s invA gene; (3) if the anthelminthic is given after Salmonella infection, then enhanced colonization of the bacteria persists; and, (4) there appears to be a ‘physical’ interaction/association between the worm and the bacteria. 

These are all robust observations. Therein lies the concern with this manuscript. The points the manuscript makes are well-taken, but the authors make no attempt at defining the mechanism underlying this helminth-bacteria interaction. They are clearly aware of this, because much of the discussion speculates on the mechanism(s) of this interaction. The data presented is a solid platform on which to build a mechanistic study. The manuscript in its current form does not go far enough – simple assays (worm/egg counts, bacterial culture/cfu enumeration) support some interesting observations. The impact of this research would be boosted by analyses of ‘how’ this happens, and even ruling possibilities out would be a valuable addition to the field. Finally, the authors note the host-parasite specificity of helminth-bacteria interactions and so it would be useful to change either the parasite or the bacterial pathogen, and determine the outcome of anthelminthic on bacterial colonization.

In summary, the data are convincing and there is no inherent flaw in the work, it, in my view, simply does not go far enough and really needs to be complemented by some analysis of mechanism.

Reviewer #2: Main comments are provided in the above sections. Overall, this was a rigorously-conducted study, yielding interesting information of public health relevance, however, further exploration of the requirement for invasive Salmonella in the bacterial persistence phenotype when H.polygyrus is removed after bacterial inoculation would be useful.

 Two experiments are needed (as described in previous sections):

- consistently measure fecal egg burdens in untreated vs PP treated mice to validate deworming for Figs 1 and 4

- Use of invA Salmonella for the 'after inoculation' PP-treatment

PLOS authors have the option to publish the peer review history of their article (what does this mean?). If published, this will include your full peer review and any attached files.

Reviewer #1: No

Reviewer #2: No
---

## [Decision Letter · Decision Letter 1]

8 Dec 2020

Dear Dr Reynolds,

We are pleased to inform you that your manuscript 'Impaired host resistance to Salmonella during helminth co-infection is restored by anthelmintic treatment prior to bacterial challenge' has been provisionally accepted for publication in PLOS Neglected Tropical Diseases.

Best regards,

Subash Babu

Guest Editor

Christine Petersen

Deputy Editor

Reviewer's Responses to Questions

**Key Review Criteria Required for Acceptance?**

**Methods**

-Are the objectives of the study clearly articulated with a clear testable hypothesis stated?

-Is the study design appropriate to address the stated objectives?

-Is the population clearly described and appropriate for the hypothesis being tested?

-Is the sample size sufficient to ensure adequate power to address the hypothesis being tested?

-Were correct statistical analysis used to support conclusions?

-Are there concerns about ethical or regulatory requirements being met?

Reviewer #1: (No Response)

Reviewer #2: Objectives, hypothesis and study design are clearly stated

**Results**

-Does the analysis presented match the analysis plan?

-Are the results clearly and completely presented?

-Are the figures (Tables, Images) of sufficient quality for clarity?

Reviewer #1: (No Response)

Reviewer #2: Results are clear

**Conclusions**

-Are the conclusions supported by the data presented?

-Are the limitations of analysis clearly described?

-Do the authors discuss how these data can be helpful to advance our understanding of the topic under study?

-Is public health relevance addressed?

Reviewer #1: (No Response)

Reviewer #2: Conclusions are justified

**Editorial and Data Presentation Modifications?**

Reviewer #1: (No Response)

Reviewer #2: The manuscript is clear as-is

**Summary and General Comments**

Reviewer #1: (No Response)

Reviewer #2: The authors provide a thoughtful and detailed response to my comments. Although some of the questions I raised were not fully addressed, the authors provide a valid justification and include data in the rebuttal that explains their attempts to answer my specific questions.

PLOS authors have the option to publish the peer review history of their article (what does this mean?). If published, this will include your full peer review and any attached files.

Reviewer #1: No

Reviewer #2: No

---

## [Editor Report · Acceptance letter]

17 Jan 2021

Dear Dr Reynolds,

We are delighted to inform you that your manuscript, "Impaired host resistance to *Salmonella* during helminth co-infection is restored by anthelmintic treatment prior to bacterial challenge," has been formally accepted for publication in PLOS Neglected Tropical Diseases.

Best regards,

Shaden Kamhawi

co-Editor-in-Chief

Paul Brindley

co-Editor-in-Chief
